# Effects of Experimental Agents Containing Tannic Acid or Chitosan on the Bacterial Biofilm Formation in Situ

**DOI:** 10.3390/biom10091315

**Published:** 2020-09-12

**Authors:** Anton Schestakow, Matthias Hannig

**Affiliations:** Clinic of Operative Dentistry, Periodontology and Preventive Dentistry, University Hospital, Saarland University, Building 73, 66421 Homburg/Saar, Germany; matthias.hannig@uks.eu

**Keywords:** tannic acid, chitosan, biofilm

## Abstract

Chitosan and tannic acid are known for their antibacterial properties. In the present in-situ study, their antibacterial and anti-adherent effects on biofilm formation on enamel were investigated. Six subjects carried upper jaw splints with bovine enamel specimens, allowing in-situ biofilm formation. During the two-day trial, subjects rinsed with experimental solutions that contained either chitosan, tannic acid (pH = 2.5), tannic acid (pH = 7) or hydrochloric acid. Water served as the negative and chlorhexidine as the positive control. Rinsing occurred four or five times following two different rinsing protocols to investigate both the immediate and long-lasting effects. After 48 h of intraoral exposure, the dental plaque was stained with LIVE/DEAD^®^ BacLight, and fluorescence micrographs were evaluated by using the software ImageJ. The results were verified by scanning electron microscopy. Rinsing with chitosan resulted in little immediate antibacterial and anti-adherent effects but failed to show any long-lasting effect, while rinsing with tannic acid resulted in strong immediate and long-lasting effects. Except for a slightly lower antibacterial effect, the neutral solution of tannic acid was as good as the acidic solution. Hydrochloric acid showed neither an antibacterial nor an anti-adherent effect on dental biofilm formation. Experimental solutions containing tannic acid are promising anti-biofilm agents, irrespective of the pH values of the solutions. Chitosan, on the other hand, was not able to prevent biofilm formation.

## 1. Introduction

According to the Global Burden of Disease 2015 Study, oral diseases are highly prevalent. Especially, dental caries affects 2.5 billion people worldwide [1]. Even though oral diseases are rarely life-threatening, they can result in financial burdens and lower quality of life [2,3,4]. Considering the high prevalence, the current prophylaxis seems not to be sufficient. Dental caries is a multifactorial disease that offers multiple targets where prophylaxis can take effect [5].

Biofilm formation starts with the adsorption of salivary and bacterial proteins to the dental surface, resulting in the formation of a protein-rich layer [6]. This so-called “acquired pellicle” has protective properties, but it also facilitates bacterial adherence to the tooth surface [7,8,9]. Oral bacteria adhere to the pellicle and produce extracellular polysaccharides, forming a bacterial community—termed “biofilm” or “dental plaque” [10]. According to Marsh (2000), the oral microbiome plays an active role in the host’s defense by reducing the chance of colonization by exogenous species [11]. Only a disturbance within the biofilm can lead to dental caries and cavities in the long term. When sugar is consumed in a high frequency or the saliva flow is reduced, bacteria can lower the local pH value by producing organic acids. Consequently, a bacterial shift to more cariogenic phenotypes occurs, resulting in further acidification and demineralization of the tooth [12]. Since dental plaque is essential in the progress of dental caries, the reduction of such by teeth brushing is a widely used method. However, the skill and time required for teeth brushing can exceed a person’s stamina or is difficult to perform due to mental or physical disability. In some situations, such as after an intraoral surgical treatment, teeth brushing is difficult for a short time by the restricted mouth opening. An existing orthodontic appliance also makes it difficult to achieve high oral hygiene standards by teeth brushing alone. [13]. Therefore, there is a demand for alternatives.

For instance, mouthwashes can supplement or even replace teeth brushing [14]. Nowadays, the antibacterial substance chlorhexidine is considered the gold standard of mouth-rinsing solutions [15]. Chlorhexidine was developed by Imperial Chemical Industries in the 1940s in search of an antimalaria agent [16]. The antiplaque effect of chlorhexidine was not known until 1970, when a study about a mouth rinse containing 0.2% chlorhexidine was published [17]. Chlorhexidine is a strong antiplaque agent due to its polycationic nature. Thereby, chlorhexidine can bind to negatively charged surfaces such as bacteria, resulting in an immediate bactericidal effect and a prolonged bacteriostatic effect as a result of adsorption to the pellicle-coated tooth surface [18,19,20]. Despite the outstanding performance, the usage of chlorhexidine can temporarily lead to taste irritation, burning sensation, desquamative lesion on oral mucosa and particularly extrinsic staining of teeth [20]. That is why alternatives are being searched. In the present study, tannic acid and chitosan have been examined.

Tannic acid belongs to the tannins, which are water-soluble polyphenols with a molecular weight from 500 to 3000 Da [21]. Tannins are naturally found in several plants and act as pesticides [22]. They are promising ingredients in mouth rinses, because they can interact with the dental plaque in different ways. Due to their astringent properties, they can inhibit several enzymes. By inhibiting the enzyme glycosyl-transferase, which is used by bacteria to adhere to the pellicle, tannins are able to inhibit bacterial adherence [22,23]. The effect on bacterial adherence was researched recently, where a mouthwash containing tannic acid showed an anti-adherent effect even eight hours after an initial application, speaking for a long-lasting effect [24]. Tannins alter the ultrastructure of pellicles, resulting in thicker and electron-denser pellicles, as shown with transmission electron microscopy [25,26]. At high concentrations, tannins even interact with bacterial membranes and alter cell permeability [27]. Furthermore, they are chelating agents and can withdraw metal ions, which are essential for microorganisms, thus inhibiting bacterial growth [21]. Tannins are also of general medical interest. They have an antioxidant and anti-carcinogenic effects and reduce the incidence of cardiovascular and neurodegenerative diseases [27,28]. Tannic acid is also categorized as a generally recognized as safe (GRAS) food additive [29]. So far, no cytotoxic effects on the oral mucosa are known.

The other substance examined in the present study was chitosan. Chitosan derives from chitin, a polysaccharide first extracted from mushrooms in 1811. Chitin consists of repetitive n-acetyl glucosamine units. It is biocompatible and biodegradable but insoluble in water and most organic solvents. As a result of different chemical modifications, the deacetylated derivate chitosan was developed [30]. Now, repetitive glucosamine units have positive charges in acidic solutions; thus, chitosan became soluble in aqueous acidic media and, therefore, can be used in a mouthwash and a binding agent for the negatively charged surface of bacteria that can interfere with the bacterial membrane [31]. Since antibacterial properties depend on positive charges, the antibacterial effect increases with a higher molecular weight and degree of deacetylation [32]. An in-situ study showed that treatment of the acquired pellicle with chitosan results indeed in a reduced adherence of bacteria [33]. However, the acidic pH of chitosan solutions not only leads to erosion but, like chitosan, it also imparts a positive charge to the pellicle, which may limit further binding of chitosan [25,26]. Chitosan in general is considered nontoxic, is often used as wound dressing and was recently used to treat oral mucositis [34,35].

The purpose of the present in-situ study is to examine the antibacterial and anti-adherent effects of tannic acid and chitosan. In order to examine to what extent the pH value plays a role for the tannic acid solution, a neutral solution of tannic acid, as well as hydrochloric acid, were prepared as experimental mouth-rinsing agents. Water served as a negative and chlorhexidine as a positive control. During a two-day trial of biofilm formation on bovine enamel specimens, subjects rinsed four or five times with different experimental solutions. By using two different rinsing protocols, the immediate and long-lasting effects were tested. The effects were evaluated by using fluorescence microscopy and scanning electron microscopy.

## 2. Materials and Methods

### 2.1. Subjects and Specimens

Six dental students (aged 24–30 years) agreed to participate in the present study. The subjects were nonsmokers, and they showed no signs of caries, periodontal diseases, reduced salivary flow nor diseases of the oral mucosa. In the past 6 months, no antibiotics were taken. The Medical Ethic Committee of the Medical Association of Saarland approved the study design (ID number 238/03, 2016). Bovine enamel samples were used to prepare rectangular specimens (edge length 5 mm, height 1.5 mm). They were obtained from lower incisor teeth of two-year-old cattle from the slaughterhouse in Zweibrücken, Deutschland. By using a cut-off machine, several rectangular slabs were made from the labial surfaces. They were then ground and polished with abrasive paper up to 2500 grit using a wet grinding machine. After that, the smear layer on the slabs was removed by ultrasonication with NaOCl for 3 min. Then, the samples were disinfected in isopropyl alcohol (70%) for 3 min, washed in distilled water and stored in it for 6 h prior to exposure in the oral cavity [36].

### 2.2. Pellicle and Initial Biofilm Formation and Application of Test Substances 

Four specimens were fixed with silicone impression material (PRESIDENT light body, Coltène/Whaledent GmbH + Co. KG, Langenau, Deutschland) to individual upper jaw splints (Figure 1), which were thermo-formed from methacrylate foils (DURAN^®^, Scheu Dental GmbH, Iserlohn, Deutschland). For each round, new specimens were fixed to the splints. The test procedure was the same for every subject. It consisted of several rounds, with each subject having to go through all the rounds (Figure 2). The washout phase between rounds was at least one whole day. For each round, splints were carried intraorally for 48 h, and only one test substance (Table 1) was applied 4 or 5 times. Two rinsing protocols were used, so that there were two runs per test substance. In the 1st rinsing protocol, 10 mL of the test substance was applied for 30 s after 3 min, 12 h, 24 h, 36 h and 47.5 h after inserting the splints. In the 2nd rinsing protocol, however, application occurred only after 3 min, 12 h, 24 h and 36 h. After the two-day trial, specimens were dismounted from the splints and rinsed with distilled water in order to remove nonadsorbed salivary remnants. Four specimens per protocol and subject were examined with LIVE/DEAD and SEM, each with two specimens. 

### 2.3. LIVE/DEAD^®^ BacLight™

The biofilm on the specimens was stained with the LIVE/DEAD^®^ BacLight™ Bacterial Viability Kit L7012 (Invitrogen, Molecular Probes, Eugene, OR, USA). That kit contains two stains targeting nucleic acid that differ in terms of spectral properties and the ability to penetrate bacteria. SYTO 9 (3.34 mM, 300-µL solution in dimethyl sulfoxide) is a green fluorescent stain that penetrates both bacteria with intact and damaged membranes. Propidium iodide (20 mM, 300-µL solution in dimethyl sulfoxide) is a red fluorescent stain that penetrates only bacteria with a damaged membrane. When both stains are present in the cell, propidium iodide displaces SYTO 9, resulting in green fluorescent bacteria with intact membranes and red fluorescent bacteria with damaged membranes. The biofilm was stained for 10 min with the staining solution, which contained 1 µL of each stain in 1000 µL of 0.9% NaCl solution. Then, specimens were investigated by fluorescence microscopy (Axio Imager.M2, Carl Zeiss Microscopy GmbH, Jena, Deutschland) using a fluorescein diacetate filter (Sigma, St. Louis, MO, USA) and an ethidium bromide filter (Roth, Mannheim, Deutschland). Of each specimen, six pictures were taken. The software ImageJ 1.52 (NIH, Bethesda, MD, USA) was used to determine the viability and the coverage of specimens with bacteria. Coverage is the area fraction of bacteria from the total area of the image. For that, bacteria were selected manually using the software, which calculated the number of pixels of the selected area. To determine the vitality, green and red bacteria were selected and evaluated separately. In addition to the number of selected areas, the brightness of pixels was taken into account. Therefore, bacteria lying on top of each other shone brighter and, thus, received a higher measured value. The software generated a single value for green and red bacteria. Vitality was determined by dividing the result of green bacteria by the sum of green and red bacteria.

### 2.4. Statistics

The fluorescence microscopic data were checked for normal distribution using the Shapiro-Wilk test. Despite transformation or elimination of outliers, they were not normally distributed (*p* < 0.05). Therefore, nonparametric tests were carried out. Differences to the negative control were determined by using the Friedmann test (one-tailed, *p* = 0.05). The Wilcoxon test (one-tailed, *p* = 0.05) was used to investigate the differences between both rinsing protocols. Statistical analysis was performed using the GraphPad Prism 8 software (GraphPad Software, San Diego, CA, USA).

### 2.5. Scanning Electron Microscopy

Two specimens per protocol and subject were investigated with a scanning electron microscope (XL 30 ESEM FEG, FEI Company, Eindhoven, The Netherlands). Therefore, specimens were fixed in a solution containing 2.5% glutaraldehyde and 0.1-M cacodylate buffer for at least 1 h at 4 °C. Then, specimens were washed with 0.1-M cacodylate buffer and dehydrated with an ascending series of ethanol. After that, specimens were chemically dried with hexamethyldisilazane for 30 min and air-dried overnight. Since biological samples have nonconductive properties, specimens were coated with carbon. Finally, the surfaces of samples were imaged at a magnification up to 20,000-fold, and the morphology was investigated.

## 3. Results

### 3.1. LIVE/DEAD^®^ BacLight™—Coverage

In the first rinsing protocol, rinsing with water or hydrochloric acid solution resulted in a coverage of 47–55% (Figure 3). Rinsing with chitosan reduced the coverage to 36%, while rinsing with tannic acid (pH = 2.5), tannic acid (pH = 7) or chlorhexidine reduced the coverage significantly (*p* < 0.05) to 2–11%. In the second rinsing protocol, after the application of water, hydrochloric acid solution or chitosan, the coverage was 50–66%. Rinsing with tannic acid (pH = 2.5), tannic acid (pH = 7) or chlorhexidine resulted in a coverage of 3–26%. In the second rinsing protocol, only chlorhexidine was able to reduce the coverage significantly (*p* < 0.05). Representative fluorescence micrographs are depicted in Figure 4. Differences between both rinsing protocols are not significant (*p* > 0.05).

### 3.2. LIVE/DEAD^®^ BacLight™—Viability

In the first rinsing protocol, rinsing with water or a hydrochloric acid solution resulted in a viability of 51–66% (Figure 5). For tannic acid (pH = 7) or chitosan, the viability was 35–43%, while tannic acid (pH = 2.5) or chlorhexidine reduced the viability significantly (*p* < 0.05) to 6–23%. In the second rinsing protocol, after the application of water, hydrochloric acid solution or chitosan, the viability was 56–73%. For tannic acid (pH = 2.5), the viability was reduced significantly (*p* < 0.05) to 30%, and rinsing with tannic acid (pH = 7) resulted in a viability of 39%. Chlorhexidine was able to reduce the viability significantly (*p* < 0.05) to 18%. The differences between both rinsing protocols were only significant for chlorhexidine (*p* < 0.05).

### 3.3. Scanning Electron Microscopy

After rinsing with water or hydrochloric acid solution, specimens were mostly covered with a biofilm (Figure 6). Rinsing with chitosan resulted in less biofilm. After an application of tannic acid (pH = 2.5 or 7) or chlorhexidine, specimens were covered mostly by pellicle and, if any, single bacteria colonies. The predominant species were cocci with short fimbriae and globule-like particles on their surfaces, regardless of what rinsing substance was used (Figure 7). The bacteria-free areas were covered by the pellicle, which consisted of globular aggregates 100–200 nm in size. Rinsing with chlorhexidine resulted in a slightly different structure of the pellicle, with globular agglomerates measuring 200–500 nm.

## 4. Discussion

### 4.1. Subjects, Specimens and Biofilm Formation

In the present in-situ study, different experimental solutions were tested for their antibacterial and anti-adherent effects on dental plaques when being used as a mouthwash. Considering the coverage and viability when rinsing with sterile water, chitosan showed little immediate effect and no long-lasting effects, whereas rinsing with tannic acid (both solutions) and chlorhexidine, on the other hand, showed a stronger immediate and long-lasting effect. Although effects were observed during the 48-h experimental trials, it is worth mentioning that subjects were dental students with healthy oral conditions and, therefore, may have a different predominant phenotype in their biofilm compared to patients with oral diseases [12]. So far, it is not known what role the phenotype plays in terms of resistance to antibacterial substances, so further research on patients with carious lesions or gingivitis is needed. 

The number of subjects is based on studies that have already been published using a similar method [36,37,38,39,40,41,42,43,44]. In-situ studies for the examination of intraorally formed biofilm or pellicle are very time-consuming. This applies both to the researcher for preparing the samples and evaluating them, as well as to the subjects. In total, subjects had to wear the splints 12 times for 48 h and refrain from their usual oral hygiene. Therefore, in addition to patience, reliability was also required of the subjects. Since it is difficult to convince people to participate in such a study, students who work in our laboratory were hired. Due to the low number of subjects and the use of a nonparametric test, significant results were less likely. Therefore, the clinical relevance was primarily assessed descriptively. The purpose of the present study was to find an alternative to chlorhexidine; therefore, clinical relevance depended on how close the effects were to those of chlorhexidine.

Bovine teeth were used for the specimens. Since they have many similarities to human teeth and, furthermore, can be provided in large numbers and uniform quality, they are often used to substitute human teeth for scientific research [45]. During the two-day trial, specimens were attached to splints and carried out in the upper jaw, while subjects rinsed with different substances twice a day as recommended by several manufactures [46]. However, depending on the rinsing protocol test, the substances were either applied four or five times. In the first rinsing protocol, a fifth rinsing occurred only 30 min prior to the ex-vivo examination; therefore, the immediate effects were observed. In the second rinsing protocol, on the other hand, the test substances were applied only four times, and the last rinsing occurred 12 h prior to the ex-vivo examination; therefore, long-lasting effects were observed. The method was chosen in order to show which substance has effects on the biofilm formation that fade away only shortly before the next rinse. For this purpose, substances must stay in the oral cavity a long time after application. This property is called retention or “substantivity”, which is requested for active ingredients of a mouthwash [47]. After the two-day trial, the specimens were examined with a fluorescence microscope and scanning electron microscope, since a single imaging method cannot fully reveal the complex structure of the biofilm [48].

Splints were carried intraorally for 48 h. An exposure time of 48 h was chosen for several reasons. In preliminary tests, a shorter exposure time did not produce enough bacteria to determine the vitality. If the entire specimen was covered by only a few bacteria, then the vitality of individual bacteria had a far greater impact on the result than a large representative number of bacteria. In addition, the exposure time was chosen because the vitality of a 48-h-old biofilm was independent of the localization of the specimens in the oral cavity [40]. The same applies for the thickness at different locations in the buccal region of the upper and lower jaw [49].

### 4.2. Negative Control

In the water-rinsing control, half of the specimen’s surface was covered with bacteria. This is in accordance with another study that used coverage instead of the widely used counting method [50]. The interindividual differences in the present study were high, as shown by a high standard deviation. The viability was over 60% for both protocols, which is in accordance with the literature [40]. The high number of dead bacteria can be explained as follows: At the beginning of biofilm formation, the tooth is colonized by a thin layer of bacteria that is accessible to antibacterial components in saliva. Additionally, the bacteria might be already dead, since the bacterial molecules responsible for adherence are still intact. Then, further bacteria adhere to persistent bacteria. The surface of the mature biofilm is still exposed to saliva, resulting in an uneven distribution of dead bacteria that is particularly localized in the lower and top layers of the biofilm [40,51,52]. Therefore, scanning electron micrographs show parts of the biofilm that have been just declared largely dead.

### 4.3. Positive Control

The substance chlorhexidine is considered the gold standard in chemical plaque control [15]. In the present study, chlorhexidine showed the highest immediate and long-lasting anti-adherent effect, pointing to a high substantivity thanks to binding to the pellicle [19]. These results were verified with scanning electron micrographs that showed bacteria-free specimen as well (Figure 6). In comparison to the negative control, the pellicle had a different shape. While in the negative control, the pellicle’s globular structure consisted of small spherical units representing protein aggregates, rinsing with chlorhexidine resulted in bigger units [6]. Alterations in the ultrastructure of the pellicle was also described using a transmission electron microscope [53]. It may be due to the binding of chlorhexidine to proteins in the pellicle [20]. On the other hand, spherical units could represent remnants of the bacteria. However, this is contradicted by the fact that cell damage by chlorhexidine is rarely seen in scanning electron micrographs [53,54]. In addition to the anti-adherent effect, chlorhexidine also has the highest antibacterial effect among the examined substances. The viability in the second rinsing protocol was higher than in the first one, resembling a stronger immediate antibacterial effect, which is in accordance with the observations by Jenkins et al. [19].

### 4.4. Tannic Acid

Tannic acid or polyphenols in general are known for their antibacterial and anti-adherent properties [24,28]. Tannins can precipitate and inhibit proteins, form chelate complexes and change pellicle structures [21,22,23,24]. The tannic acid used in the present study was obtained from gall apples of Quercus infectoria oak. Rinsing with a 5% solution, which had a pH of 2.5, resulted in a significantly reduced coverage in the first protocol that was in accordance with the literature [24,36,55,56,57]. Except to Hertel et al. (2017), other polyphenols than tannic acid were used, and the study designs differed from each other, but they were all able to demonstrate an anti-adherent effect. In the second protocol, however, the coverage was higher, speaking for a lower substantivity. That result must be verified by further tests due to the low number of subjects and the high standard deviation [58]. The viability was reduced to 30% or less in both rinsing protocols, which corresponds with the literature [24]. The antibacterial effect was not as strong as that of chlorhexidine, but the elimination of all oral bacteria is not desired anyway, since disruption of the resident microbiome can lead to dysbiosis [59]. In summary, tannic acid has an anti-adherent and antibacterial effect to the extent to supplement oral hygiene in the long term. The formation of extrinsic staining of the teeth could be a limiting factor [60]; however, discoloration was not observed in the present study due to the short duration of use. In a study by Radafshar et al. (2017), when subjects rinsed daily with a tea containing 1% tannins, discolorations were recorded after one week but were less present than in the chlorhexidine group [61]. Nonetheless, extrinsic tooth discoloration can be removed by professional teeth cleaning mainly for aesthetic reasons (Eriksen et al., 1978) [62]. Scanning electron micrographs confirmed the low coverage. However, no morphological alterations compared to the negative control were noticed. Cell lysis were expected, since tannins can disrupt membranes. Since tannic acid must penetrate the cell wall for this purpose, the concentration of the solution may not have been high enough [63]. In addition, the mode of action of tannic acid could differ from other tannins or polyphenols.

### 4.5. Hydrochloric Acid Solution and Neutral Solution of Tannic Acid

The hydrochloric acid solution was used in order to check to what extent the low pH value of the examined tannic acid influences the biofilm. The inorganic acid was chosen, because, unlike some organic acid, it has so far not shown any antibacterial or chelating effects [64]. The pH value of the solution was 2.5. Considering the low number of subjects and the semiquantitative evaluation of the fluorescence microscopic images, the solution neither had an anti-adherent nor an antibacterial effect.

Since acidification alters the predominant phenotype within the biofilm, an additional tannic acid solution was prepared that had a neutral pH value instead [65,66]. Compared to the acidic tannic acid solution, rinsing with the neutral solution showed the same anti-adherent and slightly lower antibacterial effects. Considering that chelating properties are particularly responsible for the antibacterial effect of tannic acid, the result was not expected, since chelating agents are inter-alia pH-dependent [22,67]. For instance, another chelating agent that is used in dentistry, ethylenediaminetetraacetic acid (EDTA), loses its affinity for calcium ions with an increasing pH [68]. Carboxyl groups are responsible for the chelating property of EDTA. At low pH, these groups are protonated and, therefore, not available to form chelate complexes [69]. A similar effect was expected with tannic acid, of which hydroxyl groups are responsible for the chelating properties instead. It is not known to which degree these groups are dissociated at certain pH values and which role dissociation might play. Together with the results of the hydrochloric acid solution, it is suggested that pH values play no major role for the antiplaque effect of tannic acid. Clinical use as a mouthwash is conceivable despite its lower antibacterial properties, since the anti-adherent effect is like that of chlorhexidine. So far, no in-situ studies of a neutral tannic acid solution are known.

### 4.6. Chitosan

The substance chitosan was found in 1859 and has since been used, even in many areas outside of medicine [30,70]. Thanks to many cationic groups, chitosan has an antibacterial effect and adheres to the tooth surface [32,71]. The antibacterial effect increases with the degree of deacetylation and molecular weight, which is why a high-molecular chitosan with a degree of deacetylation of over 92% was used [72]. The concentration of the acidic aqueous solution was 0.5%, because otherwise, the viscosity increases and limits the practical application as a mouthwash. Rinsing with chitosan resulted in a little immediate anti-adherent effect but failed to show a long-lasting anti-adherent effect. A comparison with the literature is difficult, since various derivatives, as well as processing and application forms, of chitosan are used. In a clinical study by Bae et al. (2006), the plaque index was significantly reduced by rinsing with a water-soluble derivative. However, the effect was only half as strong as that of a 0.1% chlorhexidine solution [73]. Besides the low anti-adherent effect, the antibacterial effect was low as well in the present study. In contrast, two clinical studies showed a significant reduction of biofilm viability by rinsing with water-soluble derivatives of chitosan [73,74]. This was probably due to the fact that the number of subjects was higher and the standard deviation was lower, and a parametric statistical test (Appendix A) was used. Despite the significance, the reduction in viability was small in both studies and the clinical relevance questionable. Overall, chitosan has minimal, if any, immediate antibacterial and anti-adherent effects. Thus, although it is used successfully in other medical disciplines, according to the results, chitosan has no use in biofilm control [31].

## 5. Conclusions

In conclusion, experimental solutions containing chitosan have only a little impact on 48-h dental plaque formation, while rinsing with tannic acid, regardless of whether an acidic or neutral solution is used, showed a very good anti-adherent effect, which comes close to the effect of chlorhexidine and is, therefore, clinically relevant. With the exception of discoloration, tannic acid does not have any of the other side effects of chlorhexidine and might be, therefore, a potential alternative to control biofilms.

## Figures and Tables

**Figure 1 biomolecules-10-01315-f001:**
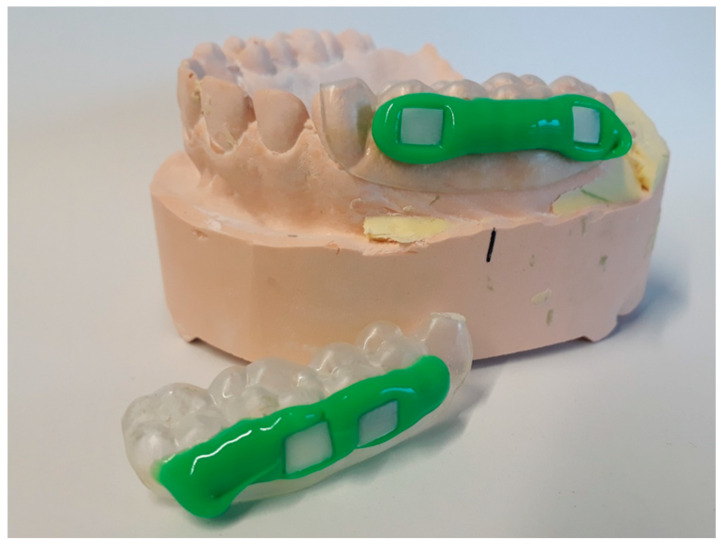
Specimens fixed to splints. Four specimens were fixed with silicone impression material to individual upper jaw splints. After the two-day trial, specimens were examined with fluorescence and scanning electron microscope, each with two specimens.

**Figure 2 biomolecules-10-01315-f002:**
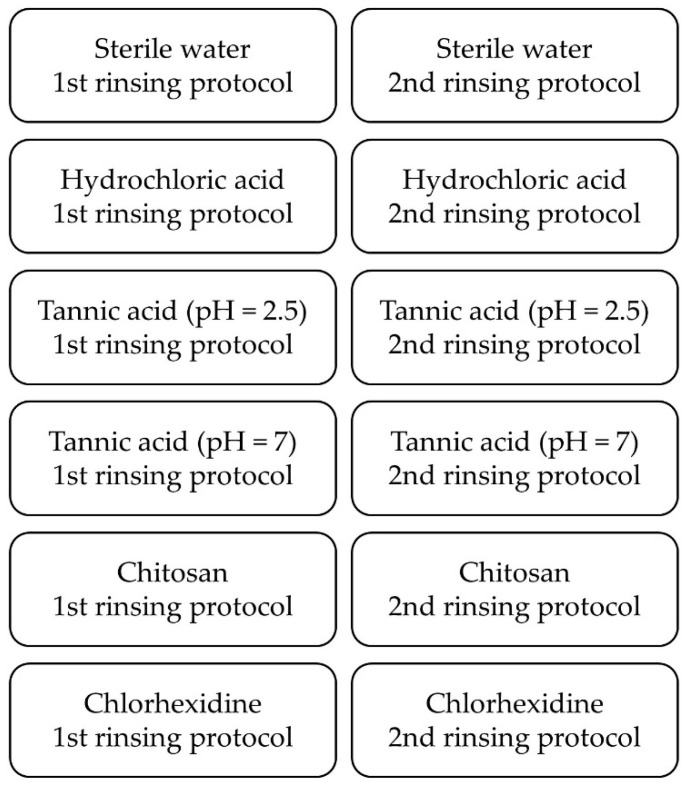
Rinsing sequences every subject has to go through. In each round, one test substance was applied according to one rinsing protocol.

**Figure 3 biomolecules-10-01315-f003:**
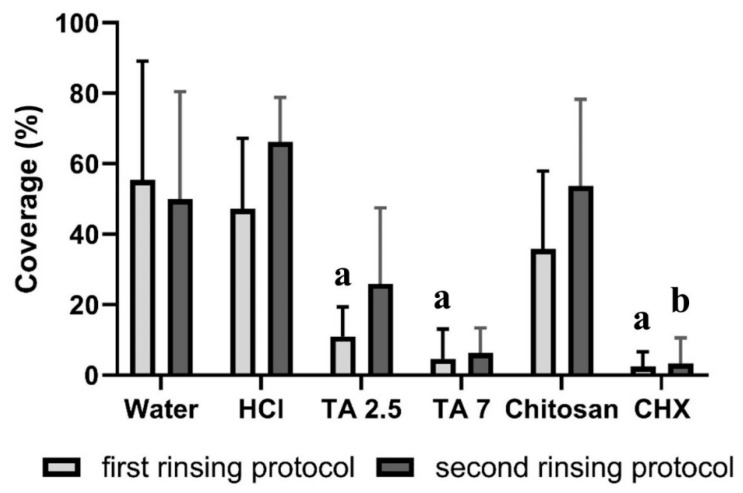
Coverage of specimens with bacteria in (%). The height of the bars corresponds to the mean value, and the line applied corresponds to ± the standard deviation. During two days of biofilm formation on bovine enamel specimens in situ, subjects rinsed 4 or 5 times with different experimental solutions. By using two different rinsing protocols, the immediate and long-lasting effects were tested. For the ex-vivo examination with a fluorescence microscope, the biofilm was stained with LIVE/DEAD^®^ BacLight™. Friedmann test: *p* < 0.05. Mouthwashes that differ significantly from water are marked with a for the first and b for the second protocol. HCl = hydrochloric acid, TA 2.5 = tannic acid (pH = 2.5), TA 7 = tannic acid (pH = 7) and CHX = chlorhexidine.

**Figure 4 biomolecules-10-01315-f004:**
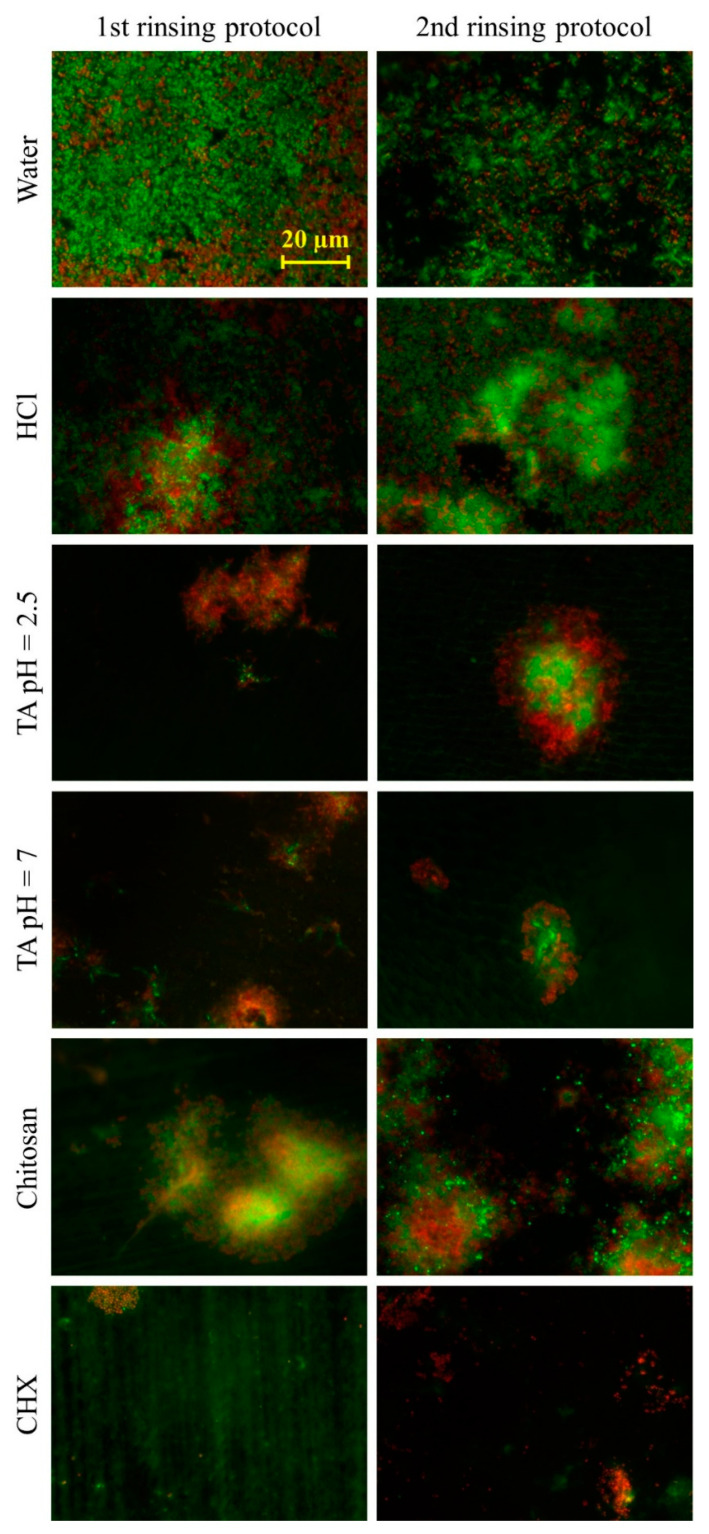
Representative LIVE/DEAD^®^ BacLight™ images. During two days of biofilm formation on bovine enamel specimens in situ, subjects rinsed 4 or 5 times with different experimental solutions. By using two different rinsing protocols, the immediate and long-lasting effects were tested. For the ex-vivo examination with a fluorescence microscope, the biofilm was stained with LIVE/DEAD^®^ BacLight™. Living bacteria are fluorescent green, and dead bacteria are fluorescent red. HCl = hydrochloric acid, TA 2.5 = tannic acid (pH = 2.5), TA 7 = tannic acid (pH = 7) and CHX = chlorhexidine.

**Figure 5 biomolecules-10-01315-f005:**
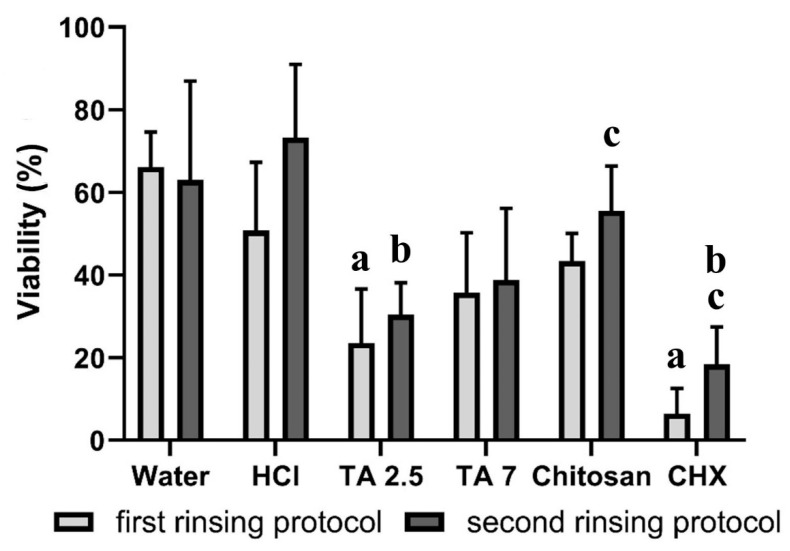
Viability of bacteria in (%). The height of the bars corresponds to the mean value, and the line applied corresponds to ± the standard deviation. During two days of biofilm formation on bovine enamel specimens in situ, the subjects rinsed 4 or 5 times with different experimental solutions. By using two different rinsing protocols, the immediate and long-lasting effects were tested. For the ex-vivo examination with a fluorescence microscope, the biofilm was stained with LIVE/DEAD^®^ BacLight™. Friedmann test: *p* < 0.05. Mouthwashes that differ significantly from water are marked with a for the first and b for the second protocol. Wilcoxon Test: *p* < 0.05. Significant differences between both rinsing protocols are marked with c. HCl = hydrochloric acid, TA 2.5 = tannic acid (pH = 2.5), TA 7 = tannic acid (pH = 7) and CHX = chlorhexidine.

**Figure 6 biomolecules-10-01315-f006:**
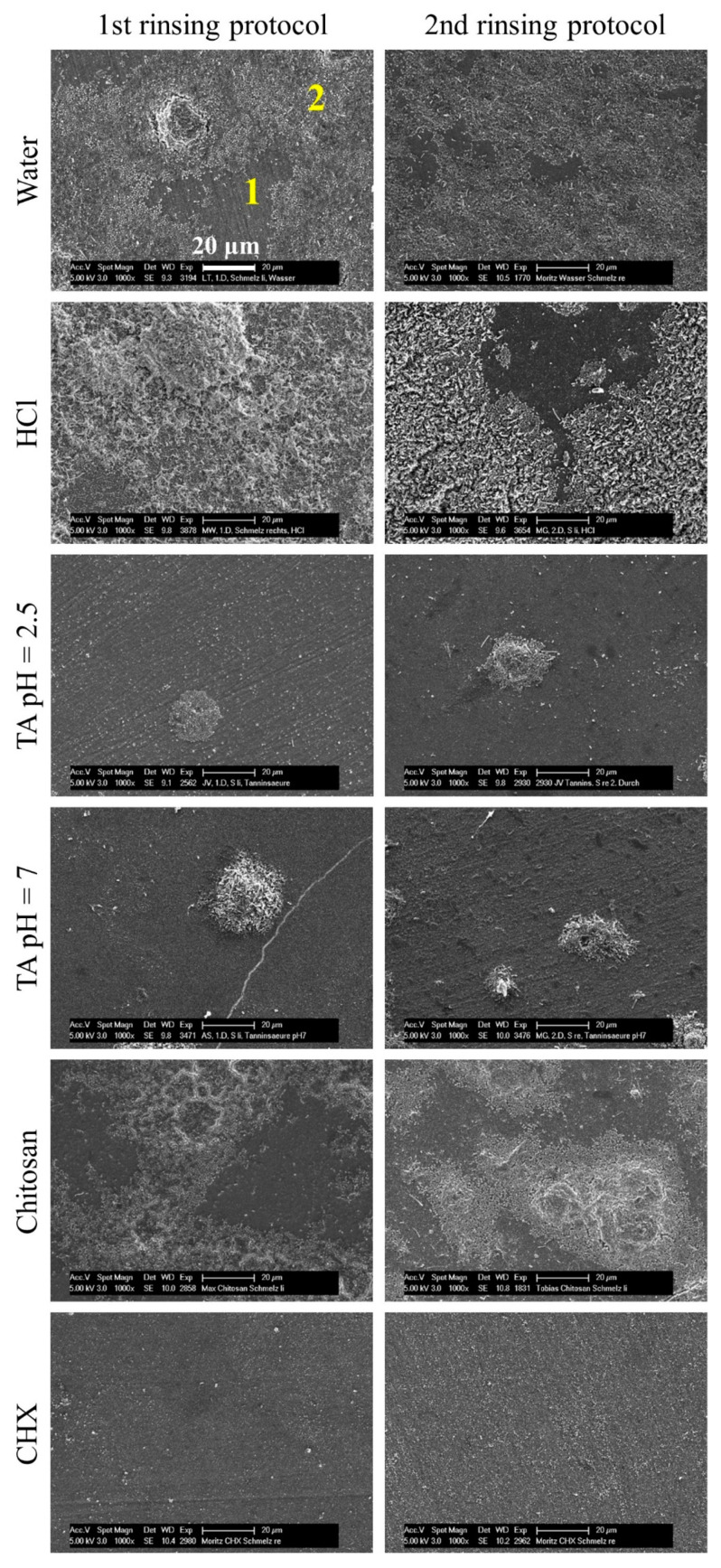
Representative scanning electron micrographs of specimens in 1000-fold magnification. Bovine enamel specimens were attached to upper jaw splints that were carried by subjects (*n* = 6) for 48 h. In the first protocol, rinsing occurred 5 times and, in the second protocol, 4 times with different experimental solutions. Micrographs show specimens covered by a biofilm that consists of the pellicle (1) and bacteria (2). HCl = hydrochloric acid, TA 2.5 = tannic acid (pH = 2.5), TA 7 = tannic acid (pH = 7) and CHX = chlorhexidine.

**Figure 7 biomolecules-10-01315-f007:**
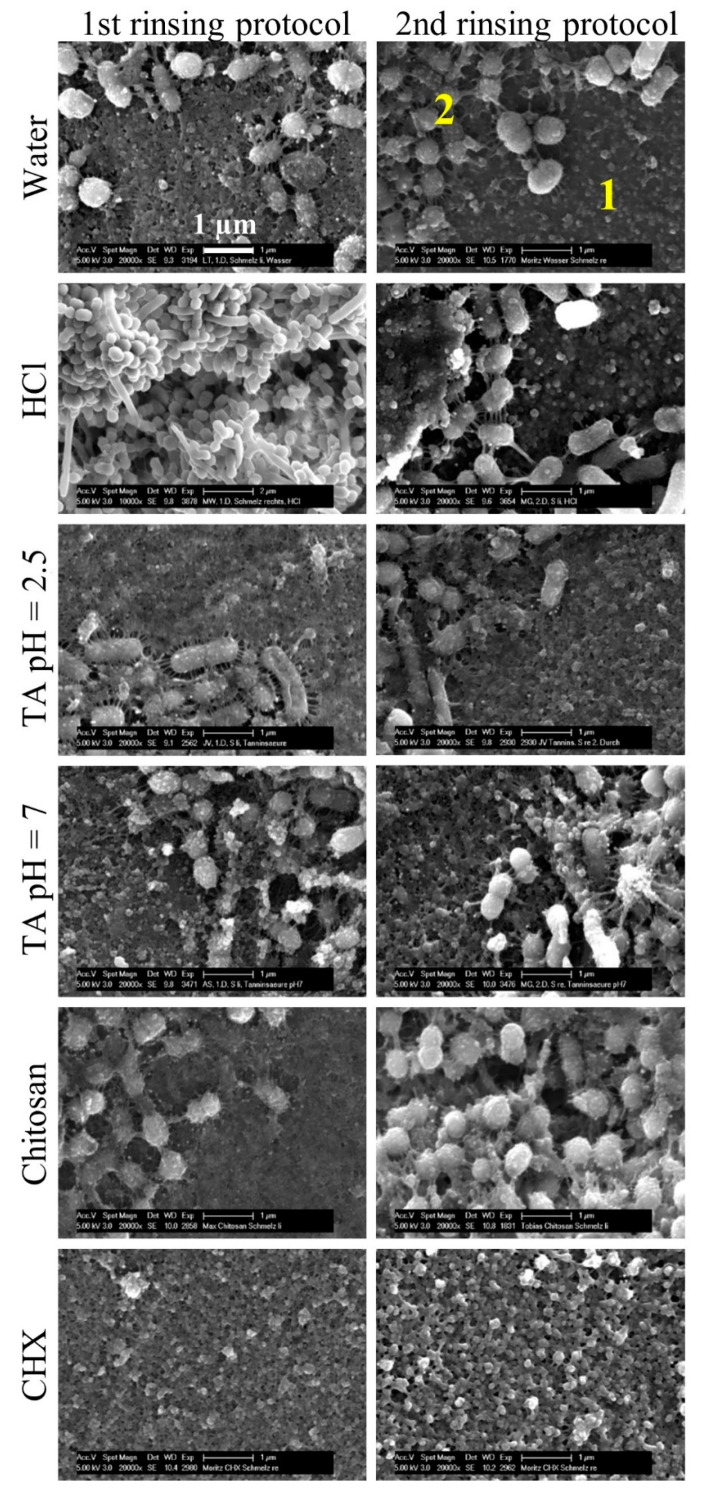
Representative scanning electron micrographs of specimens in 20,000-fold magnification. Bovine enamel specimens were attached to upper jaw splints that were carried by subjects (*n* = 6) for 48 h. In the first protocol, rinsing occurred 5 times and, in the second protocol, 4 times with different experimental solutions. Micrographs show specimens covered by a biofilm that consists of the pellicle (1) and bacteria (2). HCl = hydrochloric acid, TA 2.5 = tannic acid (pH = 2.5), TA 7 = tannic acid (pH = 7) and CHX = chlorhexidine.

**Table 1 biomolecules-10-01315-t001:** Test substances used as mouthwashes in the present study.

Test Substance	Manufacturer	Preparation
Sterile water	Ampuwa^®^, Fresenius Kabi, Bad Homburg, Deutschland	-
Hydrochloric acid	Hydrochloric acid fuming 37%, Emsure^®^, Merck KGaA, Darmstadt, Deutschland	Sterile water was titrated with hydrochloric acid to a pH = 2.5
Tannic acid, pH = 2.5	Tannic Acid, Sigma^®^, Saint Louis, MO, USA	Sterile water was added to 5 g of tannic acid to get a 100 mL solution
Tannic acid, pH = 7	Tannic Acid, Sigma^®^, Saint Louis, MO, USA	As above; additionally, the solution was titrated with sodium hydroxide to pH = 7
Chitosan (degree of deacetylation ≥ 92.6%, molecular weight 300–700 kDa), pH = 4.3	Chitosan 95/3000, Heppe Medical Chitosan GmbH, Halle, Deutschland	Chitosan was dissolved in acetic acid and sterile water; the final solution had a concentration of 5 g/L
Chlorhexidine 0.2% (Chlorhexidine-digluconate in distilled water)	Chlorhexidin 0.2 %-digluconate Lösung, Apotheke des Universitätsklinikums des Saarlandes, Homburg, Deutschland	-

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
