# Peer review of "Effects of Experimental Agents Containing Tannic Acid or Chitosan on the Bacterial Biofilm Formation in Situ"

_biomolecules, 2020, doi:10.3390/biom10091315_

Round 1

Reviewer 1 Report

Minor amendments needed:

Line 34: reference 6 formatting.

Line 37: "microflora" should not be used, as this term is now obsolete, replace with either microbiota, microbiome or microbial community(ies).

Line 44: " incompliant or impaired people". Please rephrase, and be more specific.

Lines 58-81: the authors talk about tannic acid and chitosan and how these compounds may have antibacterial properties, and affect bacterial cells. They do not provide any information about host cytotoxicity. Such information would be valuable.

Line 80: " acidic pH of chitosan solutions ". Please specify what pH values. And therefore, since chitosan may lead to erosion, why would they be considered in this study? Later in section 4.6, there is mention again of acidity (line 324), but no discussion about how this might affect erosion, even though it might be good for biofilm disruption. 

Lines 108-109: The sentence is unclear and confusing, especially the "whereas" leads to think that there were several groups of patients, those just wearing the splints and those doing mouthwashing. Please rewrite/ clarify.

Lines 104-114: A diagram/figure of the rinsing protocols might be useful to the readership. Or at least a better and clearer description in the methods section would be appreciated, because it is currently not very clear at all.

Line 134: the authors should specify in more detail the "coverage of specimens". Did the authors estimate the surface areas, if so can this information be added?

Figure 2 legend: how did the authors decide on significant differences described as a and b? Were those based on statistics or visual observations? Please specify.

Line 252: the sentence is grammatically incorrect, please rephrase. Gold standard of what?

Lines 266... Tannic acid: what about staining properties? If yes, would that be a problem in dentistry?

Line 278: replace "flora".

Line 282: " Lytic cells ". Do you mean cell lysis?

Further details on the limitations of the work should be added. For example, the low sample size used, and the methods used for measuring viable and non-viable biofilms/microbial load. Consequently, how significant are the results? The short time used for biofilm formation? (further explanations should be added on the reasons for carrying out the experiments for 48h). What further work should be done to improve the current results? (improve the significance of the work?). How clinically significant are the data presented?

Author Response

Please find author's reply to reviewer 1 comments attached: 

Reviewer 2 Report

Although the research line is interesting, the methods are not correctly explained and the results are not well presented, the microscopy figures lack quality.

Although the research line is interesting, the methods aren´t corrected explained

Page 3, lines 108-112. The rinsing protocols have to be better explained in material and methods. 

You must explain how the subjects have rinsed: for example, first with water and after 48 hours, with hydrochloric acid.

Page 3, line 108: Why were the splints in the oral cavity for 48 hours?

Page 4, line 129-130, why have you added 0.9% NaCl to the stain solution? In the BacLight protocol, the NaCl solution is used in culture conditions and preparation of bacterial suspensions, but is it necessary for the biofilm staining?

Figure 3, the CLSM images after 2nd rinsing protocol have poor quality, why?

Figure 5, in the image of the 1st rinsing protocol with hydrochloric acid, why are more biofilm than the 2nd rinsing protocol?

The authors have discussed that the high standar desviation in the experiments was due to the low number of subjects, why the experiments were not done with a larger number of subjects? 

Author Response

Please find attached author's reply to reviewer 2: 

Round 2

Reviewer 1 Report

This reviewer's remarks on including limitations of the study and statements of significance and impact of the work have been answered, but the authors did not include them in the manuscript.

Author Response

Dear Reviewer,

we thank you very much for considering our manuscript for potential publication as a research article in the journal 'biomolecules'. You suggested minor revision and reviewer 2 recommends the acceptance of the article after corrections and improvements are made. The revised manuscript was revised and corrected once more to address your comments. The explanations are attached to the file 'reply to reviewer 1'.

Thank you in advance,

sincerely

Anton Schestakow and Prof. Dr. Matthias Hannig

Reviewer 2 Report

After the corrections and improvements made by the authors, I recommend the acceptance of the article

Author Response

Dear Reviewer,

we thank you very much for considering our manuscript for potential publication as a research article in the journal 'biomolecules'. Reviewer 1 suggested minor revision and you recommend the acceptance of the article after corrections and improvements are made. The revised manuscript was revised and corrected once more to address the comments. The explanations are attached to the file 'reply to reviewer 2'.

Thank you in advance,

sincerely

Anton Schestakow and Prof. Dr. Matthias Hannig
